# Comparative Study of the Efficacy of Hyaluronic Acid, Dry Needling and Combined Treatment in Patellar Osteoarthritis—Single-Blind Randomized Clinical Trial

**DOI:** 10.3390/ijerph191710912

**Published:** 2022-09-01

**Authors:** Jorge Velázquez Saornil, Zacarías Sánchez Milá, Angélica M. Campón Chekroun, Luis Baraja Vegas, Juan Vicente Mampel, Raúl Frutos Llanes, José Manuel Barragán Casas, David Rodríguez-Sanz

**Affiliations:** 1Departamento de Fisioterapia, Facultad de Ciencias de la Salud, Universidad Católica de Ávila, C/Canteros s/n, 05005 Ávila, Spain; 2Campus San Jerónimo Guadalupe, Doctorando en Universidad Católica de Murcia, 30107 Murcia, Spain; 3Departamento de Fisioterapia, Facultad de Medicina y Ciencias de la Salud, Universidad Católica de Valencia, C/Quevedo, 2, 46001 Valencia, Spain; 4Facultad de Enfermería, Fisioterapia y Podología, Universidad Complutense de Madrid, Avenida Séneca, 2, 28040 Madrid, Spain

**Keywords:** knee osteoarthritis, myofascial trigger point, vastus medialis, anterior knee pain syndrome, range of motion, acid, hyaluronic

## Abstract

**Background:** Osteoarthritis of the knee is one of the most common ailments worldwide, and pain management of this condition is critical. **Methods:** A multicentre randomized controlled trial RCT with three months of follow-up, conducted in parallel groups: hyaluronic acid (HA), dry needling (DN) and ultrasound (US) and isometrics of quadriceps. 60 participants took part in the RCT who were diagnosed with osteoarthritis (Grade 3) of the knee by MRI and active adults (age: 23.41 ± 1.68 years; height: 1.79 ± 0.08 m; body mass: 78.33 ± 9.03 kg; body mass index (BMI): 24.14 ± 1.45 kg/m^2^). After the assigned intervention, VAS, WOMAC, IPAQ and the Star Excursion Balance test were measured at baseline. At 24 h, 15 days, 30 days, 90 days and 180 days follow-up, all variables were measured again. **Results:** Comparing statistically significant differences between groups, VAS scores were significant at post-test measurement (HA vs. US + isometric and DN vs. US + isometric) at 24 h (HA vs. DN), at 15 days (HA vs. US + isometric and DN vs. US + isometric) and at 1 month (US + isometric vs. HA and US + isometric vs. DN). **Conclusions:** There is an improvement in pain intensity in knee osteoarthritis in the short term in patients undergoing DN and conventional US + isometric treatment, but in the long term the HA group shows an improvement in pain intensity. There is also a significant difference in the improvement of knee function at different phases of the study in the various intervention groups. The combination of DN and HA in clinical practice is the best option for the treatment of osteoarthritis.

## 1. Introduction

Cartilage is a dense, specialised connective tissue that forms the transitional skeleton in the embryo and persists in the adult in joints, respiratory tract, ribs and ears. In an adult subject, there are different types of cartilage that differ in their biochemical composition, molecular microstructure, biomechanical properties and functions [1]. These are hyaline cartilage, elastic cartilage and fibrocartilage [2]. Hyaline cartilage is the most abundant cartilage in the anatomy. It covers the articular surface of bones, is flexible, elastic, pearly white and opalescent.

The actual incidence and prevalence of hyaline cartilage lesions in the knee joint is unknown [3,4,5,6]. Regarding the location and typology of knee lesions, Hjelle et al. [7] performed 1000 knee arthroscopies (on 1000 individuals), finding an overall prevalence of cartilage lesions (chondral or osteochondral of any type) of 61% and 19% for focal lesions, excluding chondromalacia patella. Of these, 14% were classified as grade I, 26% as grade II, 55% as grade III and 5% as grade IV. Overall, 57% of cases were found to have an accompanying meniscal injury and 17% of cases had an anterior cruciate ligament injury. In cases where the lesion was focal, these percentages were 42% and 26%, respectively, and 12% of the subjects had a joint lesion of both structures. The most frequent location of the chondral defects in these cases of focal injury was the medial femoral condyle in 58% and the patella in 11% of the cases. Overall, 80% were simple injuries and 38% of the subjects recalled previous trauma to the knee. For focal defects, this percentage was 61% [7,8,9,10,11].

Recently, the International Cartilage Repair Society (ICRS) has proposed a new scale for assessment and description of articular cartilage surface damage, including osteochondritis dissecans, in order to create a more universal language for communication and dissemination of advances in cartilage pathology [9,10,11].

Articular cartilage is the only tissue without vascular, nerve or lymphatic supply, properties that may condition its low intrinsic healing capacity. There is no inflammatory response to tissue damage, and, therefore, there will be no macrophage invasion to phagocytose and eliminate the devitalised tissue or migration of cells with reparative capacity within the injured area. Chondral lesions do not resolve on their own and may eventually progress to osteoarthrosis [12,13,14].

The most appropriate resolution of a chondral injury should involve regeneration with tissue identical to hyaline cartilage. Simple repair involves filling with a non-identical tissue that should be able to seal the defective area with good adhesion to the subchondral bone and complete integration with the surrounding cartilage, as well as resist mechanical wear over time and gradually be included in the natural turnover of normal tissue [15,16,17].

One of the most widely used medical treatments for osteoarthritis is the injection of hyaluronic acid (HA), also called hyaluronate, which was discovered in the vitreous humour of the bovine eye in 1934 [18], plays an important role in the repair processes of wounds and skin damage [19,20].

The use of HA in the form of intra-articular injections in patients with osteoarthritis of the knee (gonarthrosis) was called viscosupplementation and was the first indication in traumatology and orthopaedic surgery [21,22,23,24,25,26,27,28,29,30]. The purpose of this treatment is to achieve a lubricating, mechanical and biochemical effect on the joint affected by osteoarthritis, resulting in at least partial relief of painful symptoms and improved function. The effect is usually not immediate but long-term.

The physiotherapeutic treatment of osteoarthritis may include dry needling of myofascial trigger points (MTrPs), which aims to reflexively relax the target musculature. This causes the central nervous system to begin a process of regeneration of this damaged muscle so that more nutrients arrive, and if the puncture has been successful, the muscle relaxes completely and symptoms, such as local and radiating pain and muscle shortening, disappear [31,32,33,34,35,36,37] in the short term.

The aim is to test the efficacy of three possible treatments for anterior knee pain caused by grade III osteoarthritis of the knee in the short and medium term, by measuring different variables. This grade of osteoarthritis was chosen because it is the most common and the one that patients report most discomfort and inability to carry out daily activities at a clinical level.

## 2. Materials and Methods

### 2.1. Study Desing

This is a double-controlled, multicentre RCT with three months of follow-up, conducted in parallel groups between October 2020 and April 2021. The patients were treated in two centres: Hospital Santa Teresa de Ávila and FisioSalud Ávila. The study adhered to the CONSORT guidelines. It was performed in accordance with the Declaration of Helsinki and approval was obtained from the ethics committee of the Complejo Asistencial Hospital Nuestra Señora de Sonsoles under the number GASAV/2019/11. In addition, the study was registered in clinicaltrial.gov with ID NCT03743818. All subjects signed an informed consent form before inclusion.

### 2.2. Participants

A single-blind RCT was conducted with a total of 60 patients, following the CONSORT guidelines, flow chart and checklist. The sample was collected through an informative form for participation in a research study. The participants were subjects diagnosed with osteoarthritis (Grade 3) of the knee through magnetic resonance imaging and active adults because they perform some kind of physical activity that has been assessed in METs (age: 23.41 ± 1.68 years; height: 1.79 ± 0.08 m; body mass: 78.33 ± 9.03 kg; body mass index (BMI): 24.14 ± 1.45 kg/m^2^). Subjects were patients recruited by the trauma physician of the Hospital Santa Teresa (Ávila) by means of an initial screening to select and diagnose participants who met the following inclusion criteria: (i) initial medical assessment together with MRI imaging tests with a diagnosis of grade III osteoarthritis of the knee; (ii) patients aged between 18 and 55 years; (iii) patients who agree to take part in the study and sign the informed consent form. Exclusion criteria were: (i) needle phobia, (ii) neurological, vascular or other musculoskeletal pathology of the lower extremities; (iii) previous experience with DN techniques, (iv) diagnoses, such as local infection, bleeding disorders, immunosuppression and autoimmune disease; (v) lower limb dysmetria equal to or greater than 0.50 cm. All participants were asked to maintain their daily and pharmacological habits during the study.

### 2.3. Randomization and Blinding

During the first session after they signed the informed consent forms, all participants were randomized to ensure proper blinding. Each participant received the assigned intervention (i.e., DN, HA or combined ultrasound and isometric treatment). All participants were randomized among participants to control the sequential effect of the order and transfer through by an independent investigator using Epidat 3.1 (www.sergas.es (accessed on 1 September 2020)). Participants were blinded to the assigned intervention; furthermore, all participants were ingenuous to the use DN or acupuncture procedure. Moreover, researchers were also blinded, since the researcher involved in data acquisition was different to the one performing the intervention.

### 2.4. Outcome Measurement

#### 2.4.1. Part 1

Immediately after the assigned intervention, visual analogue scale (VAS), Western Ontario and McMaster Universities Osteoarthritis Index (WOMAC), Star Excursion Balanced test and International Physical Activity Questionnaire (IPAQ) were measure at baseline. At 24 h, 15 days, 30 days, 90 days and 180 days follow-up, all variables were measured again.

#### 2.4.2. Part 2

The primary outcomes were function and pain intensity in knee osteoarthritis. All dependent variables were measured in all periods.

#### 2.4.3. Visual Analog Scale (VAS)

The VAS of 100 mm length was used. This scale went from 0 (i.e., absence of pain) to 100 (i.e., maximum pain) to assess pain intensity [35]. Subjects were asked to mark the point corresponding the original pain suffered by osteoarthritis. VAS has shown high reliability for acute pain (intraclass correlation coefficient = 0.97; 95% confidence interval 0.96–0.98) [38].

#### 2.4.4. The Western Ontario and McMaster Universities Osteoarthritis Index (WOMAC)

The WOMAC was used to characterize the pain and function of the included subjects [36]; it is a self-reported, lower extremity specific questionnaire and contains 24 questions: 17 on physical function, 5 type of pain and 2 on stiffness. Each query has five answer choices varying from 0 (no, without difficulty or no symptom) to 4 (unable to engage in activities or extreme symptoms). The Spanish version of WOMAC is a valid, reliable and responsive instrument in patients with hip or knee osteoarthritis. Internal consistency yielded a Cronbach’s alpha ranging from 0.81 to 0.93 [39].

#### 2.4.5. The Secondary Outcomes Were

The Star Excursion Balance test has been used to measure this variable. It is a common clinical test to assess balance, stability and neuromuscular control of the trunk and lower limbs for the purpose of injury prevention and rehabilitation [40].

There are different methods of developing the test, but the one described by Kinzey and Armstrong [40] has been used because it is one of the simplest to perform when measuring distances by the examiner and because it is easy for the patient to assimilate, although they found moderate intra-rater reliability (<0.70) if at least six circuits of five attempts were not performed [41,42].

The test setup includes two groups of perpendicular lines: one group is vertical and horizontal lines, and the other is lines that cut at 45° to each other, all starting from the same intersection. At the intersection of the 4 lines, a box is drawn, where the subject stands at the beginning of the test with both feet inside. The subject must then move as far as possible with one of the feet towards one of the four diagonal directions, making sure that stability was achieved with adequate neuromuscular control of the supporting leg; this is why he/she cannot carry weight with the reaching leg while it reaches as far as possible over the line. The subject must return to a central position after each reach. The furthest point reached is marked and the distance from the centre of the box is measured. The test is performed five times for each direction and with each leg, resting sufficiently between each attempt. The examiner shall manually measure the distance from the centre of the asterisk to the point of reach with a tape measure in cm [41].

At the end, the average of the five attempts is taken for each direction. To adjust the result to the leg length, the result is divided by 8 times the subject’s leg length and multiplied by 100 [40]. In the present study, we performed this complete test on all patients, as mentioned above, all runs were performed five times in order to avoid bias in the measurements.

#### 2.4.6. IPAQ Scale

Implementation of the IPAQ, which began in Geneva in 1998, has been validated in several studies in European, Asian, Australian, African and American populations, with some encouraging results. The IPAQ emerged as a response to the need to create a standardised questionnaire for population-based studies worldwide, which would buffer the excess of uncontrolled information, resulting from the excessive application of assessment questionnaires that have made it difficult to compare results and the insufficiency to assess physical activity from different settings [43].

Weekly physical activity is measured by recording METs-min-week. The reference METs values are:For walking: 3.3 METs.For moderate physical activity: 4 METs.For vigorous physical activity: 8 METs.

### 2.5. Procedures and Intervention

Firstly, the patient is diagnosed by a traumatologist with osteoarthritis (Grade III) and the active or latent MTrPs are detected in the vastus quadriceps internus of the diagnosed knee, and then, depending on the assigned group, one of the three possible treatments is performed: (1) Dry needling. A single session is carried out in which a dry puncture is performed on the MTrPs of the vastus medialis quadriceps of the injured knee. Noting whether there was a local spasm response and if so, recording the number of spasm responses. The needle used in this clinical trial was a sterile needle without head and a guide tube of 0.32 × 40 mm in size (Ref. A1038P; DN needle 0.25 × 25 with guide; Agu-punt S.L., 08013, Barcelona, Spain). This group of patients were referred to FisioSalud Ávila. (2) Hyaluronic acid. The patient undergoes an infiltration of the acid by the orthopaedic surgeon in charge of performing the intra-articular infiltration technique in correspondence to the supero-external angle of the patella with hyaluronic acid (2 mL). It is carried out with a sterile field and sterile materials after skin antisepsis with sterile gloves and a self-adherent fenestrated monofilament field, syringes pre-filled with 18Gx needle acid. Cosmopor E dressing, a 10 × 10 cm crepe bandage. This group of patients are referred to the Hospital Santa Teresa de Ávila. (3) Combined ultrasound and isometric treatment. This treatment is carried out by a physiotherapist and following the homogeneous guidelines established from the design of the study. Ultrasound pulsating is applied to the anterior aspect of the knee with an intensity of 0.8 W/cm^2^ and a duration of 8 min with a frequency of 1 MHz, and then the patient is asked to perform 20 isometric contractions of 3 s each with a pause of 6 s between each one. This group of patients are referred to FisioSalud Ávila.

Subsequently, and regardless of the treatment assigned, the evaluation of the different variables is carried out by another physiotherapist who has been blinded to the application of the three possible treatments. This evaluation is carried out at FisioSalud Ávila.

### 2.6. Statistical Analysis

All variables were expressed as mean and standard deviation (SD). Normal assumption was assessed using Kolmogorov–Smirnov. In addition, variance homogeneity between groups was assessed using Levene’s test. To analyse the acute effect of DN on local and remote PPTs, an analysis of covariance (ANOVA) was performed, with the group (i.e., DN group, HA group and ultrasound and isometric group) as the independent variable and local and remote PPTs as the covariate. The effect size (ES) was for Cohen´s calculations. The ES was expressed as the difference of typified mean change. The ES was considered trivial (<0.20), small (0.20–0.59), moderate (0.60–1.19), large (1.20–1.99) and very large (>2.00). The Bonferroni correction was used to analyse the association between VAS, WOMAC, IPAQ and Star Excursion Balanced test. A “trivial” association considered r values < 0.25, “small” (r = 0.25–0.50), “moderate to strong” (r = 0.50–0.75) and “very strong” (r > 0.75). The significance level was set at *p* < 0.05. All analyses were performed using statistical analysis software (SPSS 24 Inc., Chicago, IL, USA).

## 3. Results

The flow chart shown in Figure 1 shows the number of patients studied in this clinical trial (*n* = 60). When comparing the three groups, no significant differences (*p* > 0.05) were found in socio-demographic characteristics, descriptive data or outcome measures at baseline, but the ANOVA between groups reveals significant changes in the post-test VAS scale (P = 0.01), VAS 24 h (P = 0.035), VAS 15 days (P = 0.01), VAS 1 month (P = 0.01) and VAS 3 months (P = 0.01); in the WOMAC scale, significant changes are found at 24 h, 15 days, 1 month and 3 months (P = 0.01) (Table 1).

Differences in results between treatment groups are shown in Table 2. The intervention groups showed statistically significant differences (*p* < 0.001) with a large effect size between measurement time points for VAS and WOMAC reductions. Comparing statistically significant differences between groups, VAS scores were significant at the post-test measurement (HA group vs. US + isometrics group and DN group vs. US + isometrics) (Table 2), at 24 h (HA group vs. DN group) (Table 3), at 15 days (HA group vs. US + isometrics group and DN group vs. US + isometrics group) (Table 4) and at 1 month (US + isometrics group vs. HA group and US + isometrics group vs. DN group) (Table 5).

Statistical analysis using post hoc among significant Bonferroni.

Likewise, a significant difference was observed in the WOMAC scale measurements at 24 h (DN group vs. HA group and US + isometrics) (Table 6), 15 days (DN group vs. HA group and US + isometrics) (Table 7), at 1 month (US + isometrics group vs. HA group and DN group) (Table 8) and at 3 months (differences between all groups) (Table 9). However, the rest of the measurement times and variables did not show statistically significant differences (*p* > 0.05).

Post hoc among significant Bonferroni.

## 4. Discussion

There is currently a high incidence of knee osteoarthritis [3,4,5,6]. Increased physical activity and its high intensity lead to a high percentage of injuries to the musculoskeletal system, with different pathologies that can be classified according to their severity, but in this study, we have focused on the observation of patellar osteoarthritis. This injury can lead to a deficit and impairment of physical functionality, which can lead to social restriction or even, in the case of a professional athlete, to economic and/or personal image restriction. Surgery does not show a clear improvement in this type of pathology [7,8].

Recovery in this type of injury has multiple factors that can influence the treatment and its evolution, those surrounding the patient themself (predisposition, mentality, body mass index, associated injuries, adverse events…) mean that the patient can reverse their situation in the short/medium term more quickly, and thus improve their quality of life [13].

The main objective of this clinical trial is to observe whether pain intensity (VAS) varies in the short and medium term in osteoarthritis of the knee, with three possible treatments: a standard treatment for the recovery of this type of injury in which a programme of muscle strengthening and electrotherapy is carried out, dry needling of the vastus medialis and hyaluronic acid.

Different clinical trials [44,45,46,47,48,49] show how the pain intensity measurement scale (VAS) varies in the short and medium term after applying the invasive technique of DN in different pathologies. Calvo et al. demonstrate how pain intensity decreases in patients over 65 years of age with non-specific shoulder pain after applying DN to the MTrP plus hyperalgesia in the infraspinatus muscle [33]. Similarly, Mayoral et al. show that DN is more effective than a placebo in the prevention of pain after total knee replacement [44]. This trial was carried out on 40 individuals who were divided into two groups, and the different variables were evaluated, and they reached the conclusion that DN improves the subjective sensation of pain after knee surgery. For this reason, the clinical trial carried out by us included 44 patients with knee osteoarthritis, given that, as demonstrated, DN improves after surgery and 40 patients (with a 10% loss of individuals) in the study by Mayoral et al. were sufficient to obtain high scientific evidence [44].

Mason et al. worked along the same lines in the clinical trial they conducted on a professional ballet dancer. In this study, they performed DN on the gastrocnemius, soleus and popliteus muscles of the right lower limb, and the results of this technique were evaluated in the short term, obtaining a decrease in pain intensity (measured by VAS scale) of 7 points. Therefore, the patient was able to dance again at a high level without physical limitation and perform the same dynamic movements as before the injury, such as running, jumping, turning without pain [34].

Likewise, Salom-Moreno et al. [48] conducted a clinical trial in which 27 patients with chronic unilateral ankle instability were treated with a combined programme of muscle strengthening, proprioception and DN on the peroneus longus lateralis muscle. The results of this study demonstrate that a combined protocol of muscle strengthening, proprioception and DN decreases pain intensity in the injured ankle up to one month after the physiotherapy session that included DN [48]. For this reason, the present study aims to demonstrate the efficacy of the combined protocol, which includes muscle strengthening and ultrasound, the efficacy of a DN technique in the TPrM plus hyperalgesic of the vastus medialis quadriceps of the operated limb and the efficacy of hyaluronic acid; they can decrease the intensity of pain in patients in the short and medium term (5 weeks). In addition, the study carried out by Velázquez et al. in which 44 patients were treated showed how DN reduces pain intensity in the short and medium term in patients who underwent anterior cruciate ligament surgery [49].

Another study looking at short-term pain intensity using DN was conducted by Huguenin et al. in a double-blind, placebo-controlled trial on 59 male athletes who were treated randomly. Some individuals were treated using DN on the buttock and other subjects were treated using placebo. Measurements of the variables were collected before treatment, immediately after, at 24 h and finally at 72 h post-treatment. The result of this trial is that patients who were treated with DN obtained an improvement in the VAS scale of the hamstring and gluteal muscles [47].

The aim of this clinical trial on patients with patellar osteoarthritis is to demonstrate the efficacy of DN in the short term, as previously described in the study by Huguenin et al. [47] and to add a longer time frame (5 weeks) to assess the effect of the groups in the medium term and to test whether DN is more effective than the combined treatment with muscle contraction and ultrasound or hyaluronic acid. The pain intensity of patients with DN treatment showed a significant increase from pre to post at which point the score decreased significantly from one measurement to the next. In patients with conventional treatment, the VAS scale remained unchanged from pre to 24 h, at which point the score decreased significantly from one measurement to the next. It is in the post-treatment where patients with DN treatment showed a higher pain intensity than patients with conservative treatment, showing no significant differences between treatments in the rest of the time measures.

The present study provides a great novelty with respect to the clinical trials carried out previously, and that is that no author is known to have investigated patients with patellar osteoarthritis carrying out three arms of action, including in the study of the intensity of pain in the short and medium term. It is clear from this analysis that patients have an increase in pain intensity in the short term and clearly improve over time. This information should be included in the DN methodology, since all treatments involving this invasive technique refer to a possible increase in post-treatment pain intensity [44,45,46,47,48,49]. Another aspect that has been assessed in the clinical trial is functionality studied using the WOMAC scale, which has been validated and translated into Spanish [39]. It has been widely tested in surgical or hospital populations and widely used in clinical trials, due to its sensitivity to change and proven validity. The studies cited above [44,47,49] used the WOMAC scale, which measures function and stiffness. In these trials, they obtained WOMAC scores that did not correlate with VAS scores [44,47,49]. Since this scale is validated for research and previous studies did not show a clear improvement in this variable, we relied on this scale for the measurement of function in the clinical trial of patients with patellar osteoarthritis.

In this randomised clinical study, the WOMAC variable was studied; its score remained unchanged between pre-treatment and 24 h, at which point the score decreased significantly from one measurement to the next, both for patients with DN treatment and patients with the other treatments exposed. It is from 24 h onwards that the score of patients treated with DN was lower than those treated conservatively. Therefore, it is clear from this study that patients treated with the invasive technique DN have a perception of improvement in terms of functionality at the end of this treatment that adds DN to their recovery.

The last dependent variable analysed was stability. To measure the results of stability, the Star Excursion Balance Test was used, performing the protocol as described by Gribble et al. [40]. Different studies previously cited [45,48,49] demonstrate an improvement of DN in the stability of the lower extremities in the short and medium term, and for this reason, this joint stability assessment scale was used.

In this randomised clinical trial of patellar osteoarthritis, the stability variable increases over time almost independently of the treatment group, so it can be deduced that DN has hardly any effect in the short and/or medium term on patients with osteoarthritis; they simply improve over time with the help of progressive muscle strengthening exercises, as detailed by Risberg in his article on rehabilitation for this type of pathology [50].

The DN technique is classified as a minimally invasive physiotherapy treatment that aims to relax the musculature after puncturing the most hyperalgesic active MTrP [48,49], in this case the vastus medialis. When the needle is introduced into the vastus internus muscle, the diameter of the needle can be up to four times the diameter of the myocytes it passes through, so that slight tissue damage can be felt in the myocytes traversed [26,28]. The focal lesion caused by this type of puncture is a laceration lesion [49] and its healing sequence usually follows these three phases:Destruction phase: characterised by the rupture and subsequent necrosis of the injured portion of the myocytes, the formation of a haematoma between the injured ends and the subsequent inflammatory reaction [50,51,52].Repair phase: this phase involves phagocytosis of the necrotic tissue, regeneration of myocytes and capillary proliferation of the injured area [50].Remodelling phase: maturation of the regenerated myocytes, contraction and reorganisation of the connective deposits and recovery of the functional capacity of the muscle [52].

It is estimated that the time required for complete recovery and muscle regeneration is 7 to 10 days [44,50,51,52,53]. For this reason, it was decided to use a single DN session in this clinical trial, so as not to produce a high level of tissue damage in the muscle to be treated, since the patient has already suffered damage to his or her joint and the healing and remodelling times of the adjacent tissue must be respected.

In another area, it should be noted that pulsed ultrasound produces non-thermal effects and is used to reduce inflammation, while continuous ultrasound causes thermal effects. Mensikova et al. [54] evaluated the effectiveness of pulsed and continuous ultrasound in patients with osteoarthritis. They worked with a control group (placebo ultrasound), pulsed ultrasound group and continuous ultrasound group. The treatment was 5 min once a day for 2 weeks. They found significant pain reduction effects in the pulsed ultrasound group [54].

On the other hand, Vaz et al. compared whether continuous and pulsed ultrasound was more effective than placebo ultrasound in patients with osteoarthritis [55]. They were divided into 3 groups: (1) continuous ultrasound (at a frequency of 1 MHz with an intensity of 1 W/cm), (2) pulsed ultrasound (at a frequency and intensity higher than 1:4 pulse ratio), and (3) placebo ultrasound with a 5 min treatment once a day for 2 weeks. The results show that there is no significant difference between ultrasound and placebo. They suggest that therapeutic ultrasound does not provide additional benefit in improving pain [55].

Pantovic et al. [56], evaluated the short-term efficacy of ultrasound in the treatment of pain, physical function and disability in patients with osteoarthritis. They were divided into two groups: (1) continuous ultrasound and (2) placebo group. A 1 MHz head with an intensity of 1 W/cm^2^ was used for 10 min. Treatment was 5 times a week for 3 weeks. Results indicated significant improvements in pain, functional activity and disability with the use of ultrasound. No adverse effects were reported during and after the treatment period.

### Limitations

This clinical trial aims to observe the efficacy of DN on the knee with osteoarthritis, but it is not compared with a control group or the use of placebo [44,47], so this study could be expanded in the future. The aim is to test the increase in functionality, stability and reduced pain intensity when rehabilitating such a knee injury. In addition to this technique, the physiotherapist, together with the other health professionals, must also plan a treatment for the patient in the short/medium term.

We are aware that there is a bias due to the surveillance or Hawthorne effect, given that the patient’s habitual behaviour can be modified by feeling observed, and the study lacks a placebo control group, given the difficulty that the dry needling technique offers in creating a “false treatment”.

To control as far as possible the bias, due to the vigilance or Hawthorne effect, an explanatory protocol of the test has been designed, with the indications that the examiner should give to the patients, so that it is always carried out under the same conditions. Likewise, the sample size was not calculated by means of a pilot study, which is a limitation for the generalisability of the results; it was carried out by means of a Cuyul-Vasquez convenience study [57]. Finally, the limitation of the time period, limited to a 90-day observation, is taken into account and a longer-term study will be considered for future works.

## 5. Conclusions

There is an improvement in pain intensity in knee osteoarthritis in the short term in patients undergoing DN and conventional US + isometric treatment, but in the long term the HA group shows an improvement in pain intensity. There is also a significant difference in the improvement of knee function at different phases of the study in the various intervention groups. It seems that the combination of DN and HA in clinical practice is the best option for the treatment of osteoarthritis, but more studies with larger sample sizes, more sessions and longer follow-ups are needed.

## Figures and Tables

**Figure 1 ijerph-19-10912-f001:**
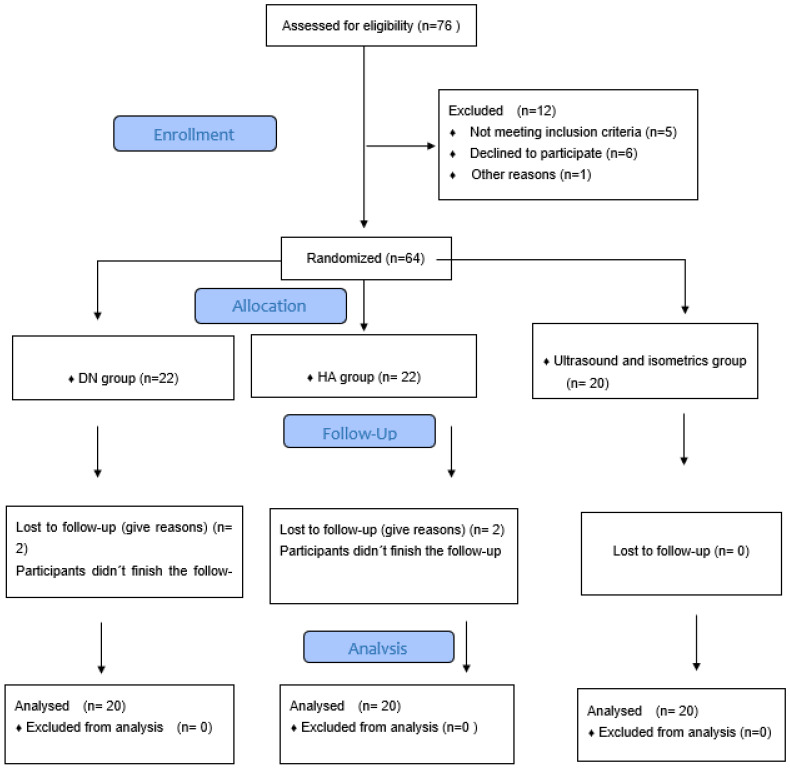
Consort 2010 flow diagram.

**Table 1 ijerph-19-10912-t001:** ANOVA.

		Addition of Squares	gl	Quadratic Mean	F	Sig.
VAS pre-test	between groups	0.233	2	0.117	0.124	0.883
	within groups	53.500	57	0.939		
	Total	53.733	59			
VAS post-test	between groups	37.300	2	18.650	18.865	0.01
	within groups	56.350	57	0.989		
	Total	93.650	59			
VAS 24 h	between groups	7.900	2	3.950	3.568	0.035
	within groups	63.100	57	1.107		
	Total	71.000	59			
VAS15 days	between groups	16.533	2	8.267	9.656	0.01
	within groups	48.800	57	0.856		
	Total	65.333	59			
VAS1 month	between groups	29.733	2	14.867	24.598	0.01
	within groups	34.450	57	0.604		
	Total	64.183	59			
VAS 3 months	between groups	136.900	2	68.450	52.761	0.01
	within groups	73.950	57	1.297		
	Total	210.850	59			
WOMAC pre-test	between groups	2.133	2	1.067	1.490	0.234
	within groups	40.800	57	0.716		
	Total	42.933	59			
WOMAC post-test	between groups	0.400	2	0.200	0.311	0.734
	within groups	36.600	57	0.642		
	Total	37.000	59			
WOMAC 24 h	between groups	35.433	2	17.717	26.061	0.01
	within groups	38.750	57	0.680		
	Total	74.183	59			
WOMAC 15 days	between groups	39.900	2	19.950	31.155	0.01
	within groups	36.500	57	0.640		
	Total	76.400	59			
WOMAC 1 month	between groups	30.400	2	15.200	30.031	0.01
	within groups	28.850	57	0.506		
	Total	59.250	59			
WOMAC 3 months	between groups	75.033	2	37.517	92.374	0.01
	within groups	23.150	57	0.406		
	Total	98.183	59			
STAR pre-test	between groups	219.024	2	109.512	0.865	0.426
	within groups	7215.889	57	126.595		
	Total	7434.913	59			
STAR post-test	between groups	293.123	2	146.562	1.240	0.297
	within groups	6737.216	57	118.197		
	Total	7030.339	59			
STAR 15 days	between groups	183.472	2	91.736	0.757	0.474
	within groups	6906.672	57	121.170		
	Total	7090.145	59			
STAR 1 month	between groups	385.645	2	192.822	1.667	0.198
	within groups	6592.457	57	115.657		
	Total	6978.102	59			
STAR 3 months	between groups	570.715	2	285.358	2.415	0.098
	within groups	6735.374	57	118.164		
	Total	7306.089	59			
IPAQ	between groups	572,210.233	2	286,105.117	0.930	0.400
	within groups	17,530,622.750	57	307,554.785		
	Total	18,102,832.983	59			
ICM	between groups	21.566	2	10.783	0.954	0.391
	within groups	644.032	57	11.299		
	Total	665.598	59			

**Table 2 ijerph-19-10912-t002:** VAS post-test. Difference between HA group and US + isometric group. Difference between DN group and US + isometric group.

VAS post-test	HA	20	7.35
	DN	20	8.00
	US + isometric group	20	6.10

**Table 3 ijerph-19-10912-t003:** VAS 24 h. Difference between HA group and DN group.

VAS 24 h	HA	20	6.85
	DN	20	6.00
	US + isometric group	20	6.65

**Table 4 ijerph-19-10912-t004:** VAS 15 days. Difference between HA group and US + isometric group. Difference between DN group and US + isometric group.

VAS 15 days	HA	20	5.60
	DN	20	4.60
	US + isometric group	20	5.80

**Table 5 ijerph-19-10912-t005:** VAS 1 month. Difference between US + isometric group with HA group. Difference between US + isometric group and DN group.

VAS 1 month	HA	20	4.55
	DN	20	3.95
	US + isometric group	20	5.65

**Table 6 ijerph-19-10912-t006:** WOMAC 24 h. Difference DN group vs. HA group and US + isometric group.

WOMAC 24 h	HA	20	6.50
	DN	20	4.80
	US + isometric group	20	6.35

**Table 7 ijerph-19-10912-t007:** WOMAC 15 days. Difference DN group vs. HA group and US + isometric group.

WOMAC 15 days	HA	20	5.90
	DN	20	4.25
	US + isometric group	20	6.05

**Table 8 ijerph-19-10912-t008:** WOMAC 1 month. Difference US + isometric group vs. HA group and DN group.

WOMAC 1 month	HA	20	4.85
	DN	20	4.65
	US + isometric group	20	6.25

**Table 9 ijerph-19-10912-t009:** WOMAC 3 months. Difference between all groups.

WOMAC 3 months	HA	20	3.90
	DN	20	4.85
	US + isometric group	20	6.60

## Data Availability

Not applicable.

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
