# Peer review of "Comparative Study of the Efficacy of Hyaluronic Acid, Dry Needling and Combined Treatment in Patellar Osteoarthritis—Single-Blind Randomized Clinical Trial"

_ijerph, 2022, doi:10.3390/ijerph191710912_

Round 1
Reviewer 1 Report
First of all, I want to note that it has been a pleasure review your manuscript. I think this is an interesting topic for clinicians who manage this prevalent condition
This study provides an insight into the efficacy of three treatments (hyaluronic acid, dry needling and combined treatment) for anterior knee pain caused by grade III osteoarthritis .
In order to improve the quality of the manuscript. After reading in depth the manuscript, I would like to make some comments and ask the authors several questions about.
- In the introducction section, when talking about the objective, it is stated that it would be at grade III. It would be useful to specify why this type has been chosen. Previously the different types were described and it is seen that the most frequent one. But it should be specified with some words to make it clearer for the reader.
- Line 97-98 … “Each included 97 subject signed informed consent. All subjects signed an informed consent before inclusion.”. It should be corrected. Repeated twice.
- Line 119 missing a parenthesis
- Review the text, the meaning of each acronym should be given the first time it appears in the text. For instance, line 100: RCT, line 128: VAS, WOMAC, IPAQ, line 111: DN….
- If you want to specify the type of study design, it is not enough not to say (RCT) at the end of the sentence where you are talking about the participants.
- In line 140 the sentence is incomplete.
- line 155: finish the sentence correctly, please.
- line 156: finish the sentence correctly.
- line 170. Delete the point after cm.
- line 185: METs values13 ¿?
- line187: 4 METs. 3. ¿?
- line 205, 206, 207.. develop it in one sentence
- line 212: finish the sentence correctly, please.
- line 225: finish the sentence correctly, please.
- line 230: delete the point after (P.0.00)
- line 230: is correct (P.0.00)?
- line 237: Table 1: Consort 2010 flow diagram. It is not a table, is a figure.
- Review the tables and their corresponding legends. It's a bit messy.
- some of the conventional treatment used (ultrasound and isometric) should have been discussed.
Author Response
Dear Reviewer:
Thank you very much for your comments, thanks to them the manuscript has been ostensibly improved. Below are the responses to your annotations, all of which have been taken into account, and I would also like to inform you that an interventional procedures manuscript subsection has been added for clarity.
Best regards and thank you for everything.

Reviewer 2 Report
Title
Title is appropriate because it is completely informative about the contents of the paper.
Abstract
The abstract respects the rules of the journal. The background and the aim are interesting. In the design is present the type of study. Methods can better explained. The clinical Impact is present but need to be better explained, increasing the number of reported references.
Text
The introduction and the discussion of the study clearly sum up the background of the study, but there are some aspects that need to be improved:
1. I would not write that dry needling is a physiotherapy practice. First because it is not true, as it is an invasive practice used mainly by medical doctors, second because the legislation on dry needling is not the same in all countries. Therefore, to be modified at line 81.
2. In the introduction, speaking of the treatment of trigger points, references should be added about the myofascial structures. The authors provide a rationale for performing the MTrPs on a review of the medical literature, but the number of references reported about this topic must be increased. For example, the various aspects of the myofascial pain can be mentioned in the article in this paragraph of the introduction, by citing the following articles:”
Dermatome and fasciatome. Stecco C, Pirri C, Fede C, Fan C, Giordani F, Stecco L, Foti C, De Caro R. Clin Anat. 2019 Oct;32(7):896-902.
Fascial Innervation: A Systematic Review of the Literature. Suarez-Rodriguez V, Fede C, Pirri C, Petrelli L, Loro-Ferrer JF, Rodriguez-Ruiz D, De Caro R, Stecco C. Int J Mol Sci. 2022 May 18;23(10):5674.
The deep fascia and its role in chronic pain and pathological conditions: A review. Kondrup F, Gaudreault N, Venne G. Clin Anat. 2022 Jul;35(5):649-659”
The methods are not clear about the procedures. There is a detailed description of the outcome measurements but not of the procedures and interventions. Add a more precisely description of them. For example, who did what, how did he do it, with a careful description so that it can be reproduced and finally the times?
The results are not reported clearly and concisely, the p-values need to be better explained. In the differences report, the p values are not present. The comparative aspects are not clear.
References
They are qualified for and updated with the lasted data. The reference list follows the format for the journal. They need to be updated about the aspects previously reported.
Tables
Different tables would better illustrate the findings.
Figures
Add figures about the procedures that highlight the key points.
Statistical Analysis
It isn’t needed further checking of data by a statistician reviewer.
General comments
The purpose of the study is original but the study needs to be improved in the introduction, methods and results. The hypothesis is defined. The methods are not clear. The study needs to be improved about the methodology. The number of references reported about the topic must be increased.
Author Response

(The authors gave the same response as above.)

Reviewer 3 Report
In this paper, Velázquez Saornil and colleagues investigated the efficacy of hyaluronic acid, dry needling and combined treatment in patellar osteoarthritis.
Overall, the paper is interesting, but there are several issues within all the manuscript’s sections, which are reported below, that need to be fixed. The English language is not so fluent; the paper needs a revision of both English language and style by a native speaker.
Abstract: Lines 17-19. a multicentre RCT with six months of follow-up, WAS conducted in parallel groups. Which are these groups? How many active adults were enrolled in the study?
Line 20: After the assigned intervention… the Authors should clearly indicate which are the interventions. Are they single or combined interventions? Otherwise, it is not clear for the reader when mentioning for instance “HA vs. US+isometric and DN vs. US+isometric”
In the abstract, there are many abbreviations that should be spelled out when used for the first time (see for instance RTC, AS, WOMAC, IPAQ, HA, US, DN and so on).
The Authors stated that VAS scores were significant at post-test measurement at 24h, 15 d, 30d. What about VAS scores at 90d and 180d? What about WOMAC, IPAQ and the Star excursion balance test?
The Introduction should be revised since it is not properly focused on the aim of the study and the rationale underlying the study.
Which are the three possible treatments? The Authors should clearly indicate which are these treatments, and why they decided to evaluate the efficacy of these treatments. Are these treatments administered alone or in combination?
Which are the main characteristics of grade III osteoarthritis? short and medium term? This should be clearly indicated in the introduction. Why did the Authors choose these time points to evaluate the efficacy of the treatments? different variables? which ones and why exactly those?
Sometimes the authors refer to “grade III osteoarthritis”, while in other lines they wrote “osteoarthritis (Grade 3)”. Please make it uniform.
The Material and Methods should be revised. How many active adults? What do the authors mean by the term of active? Are these patients the control? So are they not affected by any type of pain or disability of the knee?
The authors should specify how they were divided into study participants into the various groups. how much did they receive DN, how many HA, how many combined treatments?
Line 132: why now do the authors use the term OA which was never used before?
Which is the difference between VAS and WOMAC in pain intensity assessment? Why the authors performed both tests?
The test should be described in the order reported in line 128 (VAS, WOMAC, IPAQ and Star excursion balanced test).
IPAQ Scale: this part should be revised and explained in a clearer way. Lines 179-183, this sentence should be rewritten since it is very verbose, long and not clear for the reader. There are several typos and editing errors (see for instance: values13 are,
Lines 205-209: this part should be revised and explained in a clearer way.
The results need extensive revision. The Tables should be edited as journal guidelines. The descriptions of the tables are not clear. There are some words written in Spanish (lines 299, 302).
Line 273: ???
Lines 286-287. US+isometric (US is missing).
Lines 282-283 is written in a different way with respect to lines 286-287.
In lines 128-130, the authors stated that all the measures were performed at baseline, at 24h, 15 days, 30 days, 90 days and 180 days follow-up. Why the measures at 180days are not reported?
Where are the data regarding IPAQ and Star excursion balanced test??
Some parts of the discussion would better fit in the introduction section.
Conclusions: lines 450, 453 ND??
There are several minor issues throughout the text that need to be fixed (grammatical/typographical errors have been detected); a spell check is required throughout the manuscript to ensure that there are no other minor flaws.
Author Response

(The authors gave the same response as above.)

Round 2
Reviewer 2 Report
The authors responded satisfactorily to all my requests.Reviewer 3 Report
The authors have answered all the concerns raised by the reviewer. I have no more comments